# Anti-Wetting Performance of an Electrospun PVDF/PVP Membrane Modified by Solvothermal Treatment in Membrane Distillation

**DOI:** 10.3390/membranes13020225

**Published:** 2023-02-11

**Authors:** Yubo Xu, Long-Fei Ren, Jun Li, Chengyi Wang, Yangbo Qiu, Jiahui Shao, Yiliang He

**Affiliations:** School of Environmental Science and Engineering, Shanghai Jiao Tong University, No. 800 Dongchuan Road, Shanghai 200240, China

**Keywords:** membrane distillation, membrane wetting, electrospinning, solvothermal treatment

## Abstract

Membrane distillation (MD) is attractive for water reclamation due to the fact of its unique characteristics. However, membrane wetting becomes an obstacle to its further application. In this paper, a novel hydrophobic polyvinylidene fluoride/poly(vinyl pyrrolidone) (PVDF/PVP) membrane was fabricated by electrospinning and solvothermal treatment. The electrospun membranes prepared by electrospinning showed a multilevel interconnected nanofibrous structure. Then, a solvothermal treatment introduced the micro/nanostructure to the membrane with high roughness (*Ra* = 598 nm), thereby the water contact angle of the membrane increased to 158.3 ± 2.2°. Owing to the superior hydrophobicity, the membrane presented high resistance to wetting in both NaCl and SDS solutions. Compared to the pristine PVDF membrane, which showed wetting with a flux decline (120 min for 0.05 mM surfactant solution treatment), the prepared membrane showed outstanding stability over 600 min, even in 0.2 mM surfactant solutions. These results confirm a simple method for anti-wetting hydrophobic membrane preparation, which presented universal significance to direct contact membrane distillation (DCMD) for industrial application.

## 1. Introduction

As a necessary foundation of human society, the shortage of freshwater has been recognized as one of humanity’s biggest crises in past decades [1]. The improper discharge of industrial wastewater further deteriorates the global freshwater shortage [2]. As a result, driven by global urbanization, industrialization and population growth, there is estimated to be a 30–40% freshwater shortage in the next 10 years [3]. Therefore, water recovery from industrial wastewater and desalination of seawater are of great importance. Numerous efforts on converting wastewater/seawater to freshwater, such as multi-effect distillation, reverse osmosis and electrodialysis, have been devoted [2]. Among them, membrane distillation (MD) is an emerging hybrid thermal/membrane technology in water purification, including seawater desalination and wastewater reclamation [4]. The hydrophobic microporous membrane was commonly used between the feed stream (hot) and permeate stream (cold) to prevent the mass transfer in the liquid phase (e.g., ions, colloids and macromolecules) and enable the mass transfer in the vapor phase (e.g., water vapor) [5]. With the diffusion of vapor across the membrane driven by the transmembrane vapor pressure gradient, it would condense to liquid again on the cold permeate stream. Compared with conventional water purification processes, MD offers significant advantages including high water quality, mild operation conditions and utilization of low-grade energy [6].

The membrane, which significantly affects the mass transfer of liquid and vapor, determines the MD’s performance, including the water flux and permeate quality [3]. Polyvinylidene fluoride (PVDF), polytetrafluoroethylene (PTFE) and other low surface energy fluoropolymers are typically used for conventional MD hydrophobic membrane preparation, which usually comprises symmetric submicrometer size pores. However, membrane wetting, due to the penetration of feed into the membrane pores, is still the primary barrier for these conventional membranes in widespread industrialization, especially in treating the industrial wastewaters containing low-surface-tension molecules (e.g., ethanol) and amphiphilic substance molecules (e.g., surfactants) [7]. Once the membrane is partially wetted, the permeation flux begins to decline as the membrane pore is partly blocked. The permeated conductivity would also increase, which would lead to a lower permeation quality when the partial membrane wetting turned to full wetting [8]. Membrane wetting, a complex physical and chemical process, is mainly related with membrane properties, including surface charge and surface wettability. As MD is mainly used for wastewater with high salinity, the electrostatic interaction between membrane and foulant is negligible; therefore, membrane wetting is usually considered as the key factor guiding membrane fabrication [9].

Recently, surface wettability engineering, especially superhydrophobicity engineering, has been widely used in maintaining the non-wetting state of membrane, which ensures that the water vapor continuously permeates through the membrane pore without direct feed invasion. Increasing the membrane roughness with the re-entrant structure and decreasing the surface energy of the membrane are widely known as the main ways to achieve the surface omniphobicity [10]. In addition, electrospun membrane appears to be a popular base membrane for MD, which is attributed to its high porosity and roughness and easy modification [11,12,13]. Tang et al. [14] selected aluminum trioxide (Al_2_O_3_) nanoparticles to construct the re-entrant structure on commercial polyvinylidene fluoride (PVDF) membrane, and the resultant composite structure presented high hydroph+3+obicity (water contact angle of 159.3 ± 1.1° on the membrane surface). In our previous study [15], the in situ thermal growth of zinc oxide (ZnO) nanorods and the dip coating of low surface tension solution PDTS were applied to fabricate omniphobic FZnO-PVDF nanofibrous membrane. The as-prepared membrane exhibited high contact angles (water (164.9°) and alcohol–water (121.1°)). Although these methods are capable of achieving good MD performance, and the complexity of the membrane preparation and additional processing are still the main factors limiting their further development.

Surface roughness enhancement by fabricating multiscale nano/microstructure is an effective strategy to prepare superhydrophobic membranes. As a simple and green method, solvothermal treatment has been widely used for preparing stable roughness structures of membranes. In solvothermal treatment, the solvency/swelling effect of the solvent and the heat treatment are perfectly combined [7]. The solvent can partially dissolve the partial polymer in the membrane to form defective structures, and the heat treatment can form surface wrinkles in the polymer membrane with micro/nanostructure. Polyvinylpyrrolidone (PVP) is frequently applied in solvothermal treatment due to the fact of its excellent compatibility and perfect solubility. Lu et al. [16] successfully prepared a novel membrane with hierarchically porous ditch structures via a simply controllable solvothermal method to modify the electrospun polyacrylonitrile (PAN) nanofibrous membrane with sacrificial PVP. Ma et al. [17] successfully prepared a novel ZIF-8@PAN membrane with hierarchically rough structures through the in situ solvothermal method. The hydrophobicity and roughness of the resulting membrane were greatly improved. Therefore, the preparation of highly rough hydrophobic membranes with micro/nanostructures using solvothermal method is an effective strategy to obtain anti-wetting MD membrane.

Herein, a superhydrophobic PVDF/PVP composite membrane with micro/nanostructure was successfully prepared by combining electrospinning technology and solvothermal technology for the first time in this work. In theory, the introduction of PVP and the solvothermal reaction would endow the fibers with a rougher structure, providing the membrane with high roughness and strong hydrophobicity. Meanwhile, solvothermal treatment containing n-butanol induced the conformational flip of the PVDF molecular chain to form a membrane structure with low surface energy. Then, the membrane performance was checked by NaCl solution and the simulation of wastewater-containing typical surfactants such as sodium dodecyl sulfate (SDS). The underlying mechanism for the membrane anti-wetting property was also discussed. This study provides a potential method to prepare the anti-wetting MD membrane and also promotes the application of hydrophobic membrane in the MD industrialization.

## 2. Experimental

### 2.1. Materials

Polyvinylidene fluoride (PVDF, Mw = 700,000 g/mol) and poly (vinyl pyrrolidone) (PVP, Mw = 130,000 g/mol) were provided by Solef from Brussels, Belgium and Aladdin from Shanghai, China, respectively. Dimethylacetamide (DMAc), acetone, hydrochloric acid (HCl), n-butyl alcohol, sodium chloride (NaCl), and sodium dodecyl sulfate (SDS) were provided by Sinopharm, Shanghai, China and used without purification. All the chemicals were of ACS reagent grades. DI water with a conductivity less than 2 μS cm^−1^ was produced by Milli-Q water purification system (Millipore, Boston, MA, USA).

### 2.2. Membrane Preparation

#### 2.2.1. Electrospinning

Electrospinning solutions were prepared by dissolving PVDF and PVP in binary solvents of DMAc and acetone. The mixtures were stirred at 370 rpm and 50 °C for 12 h to obtain homogeneous solutions. The contents of the added PVP were varied from 0.0%–16.0%, and the detailed compositions of the electrospinning solutions are summarized in Table 1. Before electrospinning, the electrospinning solutions were degassed for 24 h to remove air bubbles. Then, electrospinning was conducted according to the following parameters: 0.51 mm spinneret diameter, 0.08 mm/min flow rate, 12.0 kV positive voltage, −1.0 kV negative voltage, 15.0 cm needle tip to collector, 50 rpm collector rotation, 25.0 °C temperature and 50.0% humidity. The prepared electrospun membranes were denoted as E-0, E-1, E-2, E-3 and E-4.

#### 2.2.2. Solvothermal Treatment

After electrospinning, the prepared electrospun membrane (*r* ≈ 2.0 cm) was transferred into a Teflon-sealed autoclave with a mixture of HCl (36%, 22.5 mL) and n-butyl alcohol (7.5 mL) and then treated solvothermally at 150 °C for 4 h in a vacuum oven. Afterwards, the cleaned membrane was dried at 60 °C for 12 h. Finally, the produced membranes were denoted as ES-0, ES-1, ES-2, ES-3 and ES-4. A schematic diagram of the overall membrane preparation process was shown in Figure 1.

### 2.3. Membrane Characterization

The viscosity of the electrospinning solution was tested using a viscometer (LVDV-S, Brookfield, Middleboro, MA, USA). The surface morphologies and fiber diameters of the prepared membranes were observed by scanning electron microscopy (SEM, JSM-7800F, JEOL, Tokyo, Japan). The membrane wettability was analyzed on the static contact angle measurement system (Dropmeter A-200, MAIST Vision, Ningbo, China). The pore size distribution of E-0 and ES-3 were measured by the bubble-point pressure method on a pore size analyzer (Beishide 3H-2000 PB, Beijing, China). The membrane roughness were measured by atomic force microscope (AFM, MFP-3D, Asylum Research, Santa Barbara, CA, USA). The compositions and elements of the membrane surface were determined by the X-ray diffraction analyzer (XRD, XRD-7000, Shimadzu, Tokyo, Japan). The corresponding functional groups on the membrane surface were measured by the Fourier transform infrared spectrometer (FTIR, Nicolet 6700, Thermo Fisher, Waltham, MA, USA). The membrane thermal stability was evaluated by thermogravimetric analyzer (TGA, Pyris 1, Perkinelmer, Waltham, MA, USA). The liquid entry pressures for water (LEP) of E-0 and ES-3 were measured on a lab-made experimental setup. The mechanical properties of E-3 and ES-3 were determined on an electronic universal testing machine (QJ210, Qingji, Shanghai, China). The crosshead speed was set at 10 mm/min, and the membranes were cut into rectangles with the length and width of 30 and 5 mm.

Differential scanning calorimetry (DSC) measurements were conducted using a Q200 DSC (TA Instruments, Newcastle, DE, USA) at a heating/cooling rate of 20 °C min^−1^. A more detailed description can be found in our previous studies [18,19,20]. The PVDF crystallinity was calculated from the enthalpy change during the melting process, according to the Equation (1):(1)$$X_{d}=\frac{{\Delta H}_{m}}{{\Delta H}_{0}}$$

The ∆*H_m_* (J/g) is the enthalpy of the crystal melting of the PVDF in the membrane, ∆*H_0_* = 105 J/g is the crystal melting enthalpy of the complete crystallization of the PVDF.

### 2.4. Membrane Distillation Operation

The membrane performance was evaluated using a direct contact membrane distillation (DCMD) module, as shown in Figure 2. This laboratory-scale module consists of a feed stream (1.5 L), permeate stream (0.5 L) and prepared membrane (effective area of 9.6 cm^2^). The feed stream and permeate stream were circulated at 1.0 and 0.5 L min^−1^, respectively. The temperature difference between the feed stream and permeate stream was set to 40 °C when their temperatures were maintained at 60 and 20 °C, respectively. To further evaluate the membrane’s anti-wetting performance, the feed concentrations of 35.0 g L^−1^ NaCl with/without SDS (0.05, 0.1 and 0.2 mM) were used in the experiments. The water permeation flux (*J*, L m^−2^ h^−1^) and salt rejection (*R*, %) were calculated according to our previous studies [19,20].

## 3. Results and Discussion

### 3.1. Membrane Morphology

Figure 3 shows that all of the electrospun membranes prepared from different polymer solutions exhibited a multilevel interconnected nanofibrous structure. The mean nanofiber diameter of the pristine electrospun PVDF membrane (E-0) was 199.3 nm (Figure 3A). With the addition of PVP, the nanofiber diameters began to increase to 416.4 nm (E-1, Figure 3B). When the PVP concentrations further increased from 2.0% to 16.0% in polymer solution, the conductivity also increased from 361 to 536 μS cm^−1^, facilitating the stretching of the PVDF-PVP chains [21]. Therefore, the resultant nanofiber diameters gradually decreased to 306.3–334.1 nm (E-2–E-4). Compared with the PVP-free membranes, not only did the nanofiber diameter increase but increasingly the nanofibers gradually adhered together with the PVP addition. This nanofiber adhesion phenomenon could be attributed to the high molecular weight of the PVP and the high viscosity of the polymer solution with a higher concentration of PVP. The viscosity of the electrospinning solution of E-0 was 3944 cP. After the addition of PVP, the viscosity of the electrospinning solutions of E-1 to E-4 increased to 4761, 5429, 8809 and 14,830 cP, respectively. Therefore, in subsequent membrane preparation, the appropriate amount of PVP addition should be selected.

After the solvothermal treatment, the PVDF membrane (ES-0) still showed a similar nanofibrous structure, while only a few nanofibers were partly decomposed to tiny flocs. This phenomenon was also confirmed by the nanofiber diameter measurement, as the mean diameter slightly decreased from 199.3 (E-0) to 184.0 nm (ES-0). Compared with the high stability of the PVDF, PVP was more easily removed and shaped during the solvothermal treatment. The reduction in the fiber diameter caused by the solvothermal treatment also significantly reduced the mechanical strength of the membrane. The tensile strength of ES-3 reduced to 12.1 MPa, which was lower than the 19.2 MPa for E-3. This might be due to the smaller pulling force between the finer fibers. As a hygroscopic polymer with a strong interaction with HCl and n-butyl alcohol, membrane swelling was observed during the solvothermal treatment after the PVP addition, which caused the increase in the nanofiber diameter [22]. With lower PVP concentrations (2.0–4.0%), the effect of the decomposition was more important than swelling on membranes; therefore, the nanofiber diameters still presented a decreasing tendency on ES-1 (291.4) and ES-2 (292.8) in comparison with E-1 and E-2. However, the effect of swelling was more obvious with higher PVP concentrations (8.0–16.0%), which resulted in the increased nanofiber diameters on ES-3 (329.5 nm) and ES-4 (434.4 nm). Meanwhile, it could be seen that the morphology modifications on the membranes with PVP were much more obvious. As depicted in Figure 3F–J, the nanofibers became corrugated, leading to roughed nanofibrous structures and increased surface roughness. This may be the result of dissolution of PVP in the solvent and decomposition of PVP from the membrane fibers. In theory, the newly formed roughened nanostructure of the fibers could prevent the sagging of the liquid–air interface to result in a higher water repellency [23].

### 3.2. Wettability and Roughness of Membranes

As a prerequisite, the superhydrophobic surface of a membrane should be well constructed to guarantee its high anti-wetting performance in DCMD [24]. Figure 4A evaluated the wetting behaviors of different membranes by water contact angles (WCAs). The WCA on the pristine electrospun PVDF membrane (E-0) was 138.3 ± 5.7°. PVP was usually used in tailoring membrane hydrophilicity due to the fact of its hydrophilic properties [22]; thus, the WCAs gradually decreased to 134.1 ± 3.0° (E-1), 96.6 ± 2.1° (E-2), 88.1 ± 15.1° (E-3) and 47.7 ± 18.7° (E-4), respectively, with the increase in the PVP concentrations in polymer solution from 0.0% to 2.0%, 4.0%, 8.0% and 16.0%, respectively. It was clearly concluded that the PVP dosage was negatively correlated with WCAs (*R^2^* = 0.9079, Figure 4B). The hydrophilization on the membrane surface was more obvious at high PVP concentrations on account of the appearance of C-N and C=O groups, which facilitated the formation of hydrogen bonds [25]. 

In contrast, the WCAs for ES-0 to ES-4 were 147.0 ± 3.5°, 145.1 ± 2.6°, 152.6 ± 4.8°, 158.3 ± 4.2° and 141.9 ± 3.1°, respectively, and therefore the solvothermal treatment was proved to be particularly advantageous in membrane anti-wetting improvement. As can be found in Figure 4B, the WCA variations on the different membranes presented a polynomial linear regression relationship (*R^2^* = 0.6050) in general. The WCA increase in ES-0 was largely ascribed to the tiny abrasion of electrospun PVDF nanofibers. For ES-1, ES-2 and ES-3, apart from the abrasion of PVDF during the solvothermal treatment, the decomposition of the hydrophilic PVP and the roughening of the nanofibers were also effective in improving the hydrophobicity. As confirmed previously in the SEM analysis, these effects were more obvious at the higher PVP concentrations, and the highest hydrophobization improvement was achieved on ES-3. However, the hydrophobicity of ES-4 was even lower than that of ES-3, which might be attributed to that the relatively excessive PVP in ES-4 led to the nanostructure’s collapse during membrane swelling and excessive residues of hydrophilic PVP. Figure 5 presents the morphology of water droplets on E-0, E-3 and ES-3 after 60 s. Compared to E-3, ES-3 remained unwetted, which further confirmed that the changes in the membrane WCA and surface morphology were brought by the solvothermal treatment. Therefore, in the following experiment E-0, E-3 and ES-3 were selected for further investigation of the underlying mechanism and to verify their anti-wetting performance.

The membrane surface roughness is an important parameter, which is significantly affected by the membrane surface structure, and has significant effects on the membrane wettability [14]. The AFM characterization in Figure 6 and Table 2 showed that the average roughness (*R_a_*) and root mean square surface roughness (*R_q_*) for E-0 were 336 and 435 nm, respectively. After the addition of PVP, the *R_a_* and *R_q_* for E-3 significantly increased to 556 and 727 nm. This indicated that PVP was helpful in the nanostructure construction of the fibers. With the formation of corrugated and roughed nanofibers during solvothermal treatment, the *R_a_* and *R_q_* for ES-3 further increased to 598 and 764 nm, which were approximately 1.8 times higher than that for E-0. This variation in the membrane roughness was positively correlated with the variation in the membrane intrinsic wettability. As a result, the increased roughness on the hydrophobic membrane surface would lead to the increased WCAs [26]. Table 2 also presents the porosity and pore size of E-0, E-3 and ES-3. It was shown that the porosity and pore size of E-3 were smaller than E-0, which could be attributed to its larger fiber diameter and fiber adhesion. However, the pore size and porosity of ES-3 after solvothermal treatment were significantly increased, which was due to the removal of a portion of the PVP during the solvothermal treatment.

### 3.3. Membrane Composition

#### 3.3.1. FTIR Analysis

The surface chemistry and functional groups in the membranes were investigated using FTIR analysis. As shown in Figure 7A, in the pristine PVDF membrane (E-0), the characteristic spectra of the PVDF were noted. In the range from 1000 to 1400 cm^−1^, several intense absorption peaks were observed, which were representative of fluorine compounds. For example, the absorption peak (pink block) appearing at 1175 cm^−1^ was assigned to the symmetrical stretching of the -CF_2_ vibrations [27]. In addition, both α- and β-phase PVDFs were observed in E-0, as the absorption bands of 1175, 1400 and 1430 cm^−1^ were attributed to the α-PVDF [28] and the absorption bands of 839, 878 and 1274 cm^−1^ were attributed to the β-PVDF [29]. It is noteworthy that compared with E-0, the transmittance intensity corresponding to α-PVDF in ES-3 was lower, while that corresponding to β-PVDF was higher, which indicates that ES-3 contained more β-PVDF. This might be due to the induced flipping and change of the PVDF molecular chains during the solvothermal treatment. When PVP was added, the absorption characteristic peak of PVDF was still present but slightly shifted in E-3. Meanwhile, the characteristic peaks of PVP at 1275 and 1655 cm^−1^ (yellow block) were observed, which were assigned to the stretching vibrations of C-N and C=O, respectively. These peak intensities showed that the PVP-related characteristic peaks dominated in E-3, and the PVDF-related characteristic peaks were greatly weakened. This might be due to the fact that a large amount of PVDF was wrapped by PVP in the fiber interior and could not be detected by FTIR. The FTIR spectrum of ES-3 combined the characteristic peaks of PVDF and PVP, which was confirmed by the absorption peaks at 1175 and 1655 cm^−1^. Compared with E-3, the intensity of the characteristic PVDF peaks in ES-3 were stronger and the characteristic PVP peaks in ES-3 were weaker, which indicated that the solvothermal treatment removed part of the PVP from the fiber surface. Therefore, a rougher surface profile of the fibers can form, which leads to a stronger hydrophobicity.

#### 3.3.2. XRD and DSC Analysis

In order to determine the crystal phase of the membrane samples, X-ray diffraction (XRD) measurements were performed, as shown in Figure 7B. For E-0, the small shoulder peak at 18.5° corresponded to α-PVDF, and the sharp peak at 20.1° corresponded to β-PVDF [30], which was in agreement with the previous FTIR results. Compared with E-0, the diffraction peak height and peak area of E-3 were significantly larger, indicating that the introduction of PVP led to the increase in the crystal content in the membrane, and the peak area became larger because the characteristic diffraction peaks of PVP and PVDF were partly overlapped [31]. After the solvothermal treatment, the XRD diffraction peaks of ES-3 and E-0 converged, implying that PVDF was still the main component of ES-3, but part of the PVP was removed.

The differential scanning calorimetry (Figure 7C) revealed the exothermic and endothermic properties of the membrane samples. In the DSC curve of E-0 (black curve), there was only one obvious exothermic peak at 163.98 °C, which indicates a melting point of approximately 163.98 °C for the original PVDF membrane. In the DSC curve of E-3, two peaks appeared at 83.19 and 161.15 °C, corresponding to the exothermic changes in the PVP and PVDF, respectively. This confirmed the successful addition of PVP in E-3. Moreover, the decreased melting point of PVDF in E-3 (161.15 °C) compared to E-0 (163.98 °C) indicated that the tight binding of PVP and PVDF in E-3 induced the molecular chain change of PVDF. The peak of the DSC curve of ES-3 appeared at 169.40 °C, which was much higher than that of E-0 (163.98 °C), and the peak at 83.19 °C was no longer detected; this indicated that most of the PVP was removed by solvothermal treatment, and the melting point of PVDF was increased. The crystallinity of the PVDF in E-0, E-3 and ES-3 was calculated to be 24.2%, 19.8% and 31.1%, respectively. This indicates that the addition of PVP decreased the crystallinity of the PVDF, but the solvothermal treatment promoted the crystallization of the PVDF. It is noteworthy that the results of the crystallinity coincided with the variation in the PVDF melting point. This is attributed to the fact that n-butanol promoted the conformational flip of the PVDF molecular chain during solvothermal treatment, resulting in a more stable structure of PVDF. 

#### 3.3.3. Membrane Thermal Stability

The TGA analysis was used to investigate the membrane thermal stability, and the corresponding TGA curves of E-0, E-3 and ES-3 are shown in Figure 7D. A typical two-step decomposition trend was exhibited on the TGA curve of E-0. No obvious weight loss (<3.0%) was presented before 313.9 °C, indicating that 313.9 °C was its thermal decomposition temperature (*T_d_*). The first decomposition step mainly occurred in the temperature range of 350 to 420 °C, which could be attributed to the decomposition of PVDF [32,33]. Another decomposition step took place between 420 and 700 °C due to the further degradation of carbonaceous residue [32,33]. E-3 exhibited an unexpected three-step decomposition trend. With the addition of PVP, E-3 first showed weight loss at 52.0 °C, and this may be due to the excellent water solubility of PVP which made the E-3 sample absorb water vapor from the air. The second obvious weight loss temperature of E-3 was observed between 290 and 300 °C, which was much lower than the first weight loss temperature of E-0. This phenomenon shows that the thermal stability of E-3 was worse than E-0, and it was probably inferred from the combination of PVP and PVDF during the electrospinning process, which led to the generation of an instable blend, further resulting in the decomposition process under low temperature. This was also confirmed by the lower melting point of E-3 compared to E-0, as shown in Figure 7C.

After solvothermal treatment, ES-0 showed the best thermal stability behavior as compared with E-0 and E-3. Moreover, the *T_d_* of ES-3 was detected at 414.7 °C, increasing by 32.1% and 697.5% than those of E-0 and E-3. Two major weight losses occurred between the temperatures of 430–490 and 490–700 °C, which were related with the decompositions of PVDF/PVP and carbonaceous residue, respectively. This indicated that the solvothermal treatment was beneficial in thermal stability improvement: (i) the easily degradable composition (PVP) was directly removed during this process, (ii) PVDF crystallinity increased, and (iii) PVDF molecular chain conformation flip induced by solvothermal treatment formed a more stable state.

### 3.4. Membrane Anti-Wetting Performance in DCMD

#### 3.4.1. NaCl Solution

Figure 8A presented the water flux and permeate conductivity of E-0 and ES-3 in the DCMD experiments for the NaCl solution. The pristine electrospun PVDF membrane exhibited an average water flux of 15.0 LMH. Meanwhile, the permeated conductivity gradually increased from 2.9 to 5.3 μS cm^−1^ during the first 480 min. After 480 min, the sharply increasing conductivity tendency indicated the non-wetted phase of the surface was developed into the partially wetted phase [8]. Possibly, the extended operation at high temperature may have enlarged the membrane pores of E-0 and further resulted in the reduction of its ability to resist wetting, thereby causing the wetting of partial pores. In contrast, a significant desalination improvement was observed on the membrane after solvothermal treatment. The permeate conductivity of ES-3 remained at 3.2 μS cm^−1^ until 600 min, which was essentially unchanged compared with the initial 2.0 μS cm^−1^. This implies that ES-3 had a stronger wetting resistance to the NaCl solution compared to E-0, confirming that the solvothermal treatment modification greatly improved the membrane wetting resistance.

In addition, the large pore size was able to reduce the vapor transfer resistance and promoted the vapor transfer rate, yielding the relatively high initial water flux of 21.9 LMH that was observed in ES-3, which was 38.6% higher than that of E-0. This was due to the fact that a part of the PVP was removed during the solvothermal treatment, offering more free space for water vapor to transfer across the membrane (Figure 8B). 

Although ES-3 had a larger pore size than E-0, the higher roughness and stronger hydrophobicity endowed ES-3 with higher water flux while maintaining stable desalination performance. As a result, no further conductivity increases were observed during the DCMD operation. In summary, ES-3 was a more robust MD membrane, because the solvothermal treatment brought a micro/nanostructure of fibers with significantly enhanced roughness and hydrophobicity, thus allowing the membrane to have greatly improved resistance to wetting.

#### 3.4.2. SDS Solution

To test the membrane wetting resistance to surfactant, a NaCl/SDS solution was used as the feed solution, and the water flux and permeate conductivity of the membrane were monitored by gradually increasing the SDS concentration. The initial feed solution was 3.5 wt% NaCl solution, and then SDS was added to the feed solution every 60 min up to 0.2 mM, as shown in Figure 9A. In theory, increasing the SDS concentration would reduce the surface tension of the feed solution, thus inducing membrane wetting. In the initial state, all membranes exhibited stable water flux and permeate conductivity. As we know, once the membrane wetting took place, the membrane started to lose its hydrophobicity locally, leading to continuous water bridging; as a result, the water flux began to decline and the permeated quality began to decrease. When the SDS concentration increased to 0.05 mM, the flux of E-0 started to decrease significantly, which indicates that E-0 had been wetted. As the SDS concentration further increased to 0.1 mM, the permeate conductivity of E-0 began to increase significantly and the rate of increase became more rapid. This indicated that E-0 transitioned from partial pore wetting to full wetting with the gradual accumulation of SDS on the membrane surface; thus, the salt and surfactant in the feed solution invaded into the permeate.

Interestingly, ES-3 exhibited a much more stable desalination performance even at 0.2 mM SDS, and the increase in the permeate conductivity could be negligible over the 400 min test. The excellent anti-wetting properties of ES-3 demonstrated the robustness and durability of the hydrophobic membrane in MD after solvothermal treatment. In terms of water flux, both E-0 and E-3 exhibited lower fluxes when performing desalination of the SDS solutions compared to the treating NaCl solutions, which might be attributed to the lower surface tension of the SDS solution. Overall, the solvothermal-treated PVDF/PVP membrane had greater resistance to SDS wetting than the original PVDF membrane.

For comparison, some of the previously reported MD performances of the anti-wetting membranes are summarized in Table 3. The water flux and salt rejection at a temperature difference of 40 °C of ES-3 were higher than or comparable with the other results reported, even with 0.2 mM of SDS.

### 3.5. Membrane Anti-Wetting Mechanisms

One of the main indicators for measuring membrane wettability is the liquid entry pressure (LEP), which is influenced by the surface energy of the membrane material, the surface tension of feed solution and the membrane pore size and geometry. It seemed that the membrane wetting was prone to occurrence once the ∆*P* surpassed the LEP or once excessive SDS accumulated on the membrane surface and in the membrane pores. LEP is expressed as follows [39]:$$LEP=-\frac{2B\gamma_{L}COS\theta }{r_{max}}$$
where *B* is a pore geometry coefficient, *γ_L_* (N m^−1^) is the surface tension of liquid, *θ* (^o^) is the contact angle between liquid and membrane, *r_max_* (μm) is the maximum pore size of the membrane. The LEP values of ES-3 and E-0 were tested separately, and the results are shown in Figure 9B, the LEP of ES-3 and E-0 were 124.3 ± 2.1 and 93.9 ± 1.5 kPa, respectively, and the solvothermal treatment increased the LEP of the membrane by approximately 30 kPa. Thus, ES-3 showed a better performance during the desalination of NaCl solution and SDS solution.

According to the LEP expression and experimental results in this work, the improvement in the LEP of ES-3 was mainly due to the change in the roughness from the formation of the micro/nanostructure of the membrane fibers and the significant increase in contact angle. Moreover, the DSC analysis and TGA analysis confirmed the change in the conformation of the PVDF molecular chains, which may make the surface energy of PVDF further reduced, and this may be one of the reasons for the better anti-wetting property of ES-3. These results provide strong evidence that the solvothermal treatment of ES-3 mitigated the membrane wetting by SDS solution and NaCl solution.

## 4. Conclusions

In this study, a PVDF/PVP composite membrane was prepared by solvothermal method for the first time and applied for MD to address the membrane wetting issue. The solvothermal treatment significantly enhanced the wetting resistance and hydrophobicity of the membrane. Compared with E-3, ES-3 had a higher water contact angle (158.3 ± 2.2°) and larger roughness (764 nm) due to the introduction of a micro/nanostructure. Chemical analysis showed that most of the PVP in E-3 was removed during the solvothermal treatment due to the solvent dissolution, the crystallinity of PVDF was significantly increased and the crystalline composition was changed due to the solvent induction. These endowed ES-3 with greater stability and resistance to wetting. Therefore, compared with E-0, ES-3 exhibited stronger resistance to wetting in both NaCl solution and SDS solution desalting and can stably treat SDS solutions over 0.2 mM. The higher LEP value of ES-3 compared to E-0 was the main reason for its enhanced wetting resistance. Overall, the proposed design and method for obtaining an anti-wetting membrane by solvothermal treatment showed feasibility in MD desalination applications.

## Figures and Tables

**Figure 1 membranes-13-00225-f001:**
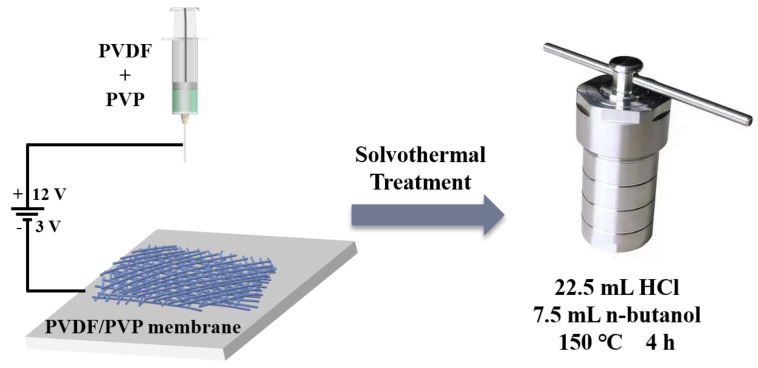
Schematic diagram of the preparation process of the PVDF/PVP membrane.

**Figure 2 membranes-13-00225-f002:**
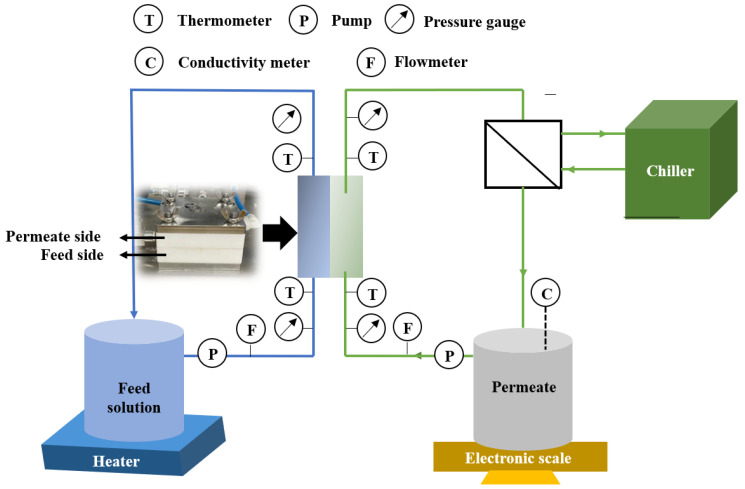
Diagram of membrane distillation (MD) setup.

**Figure 3 membranes-13-00225-f003:**
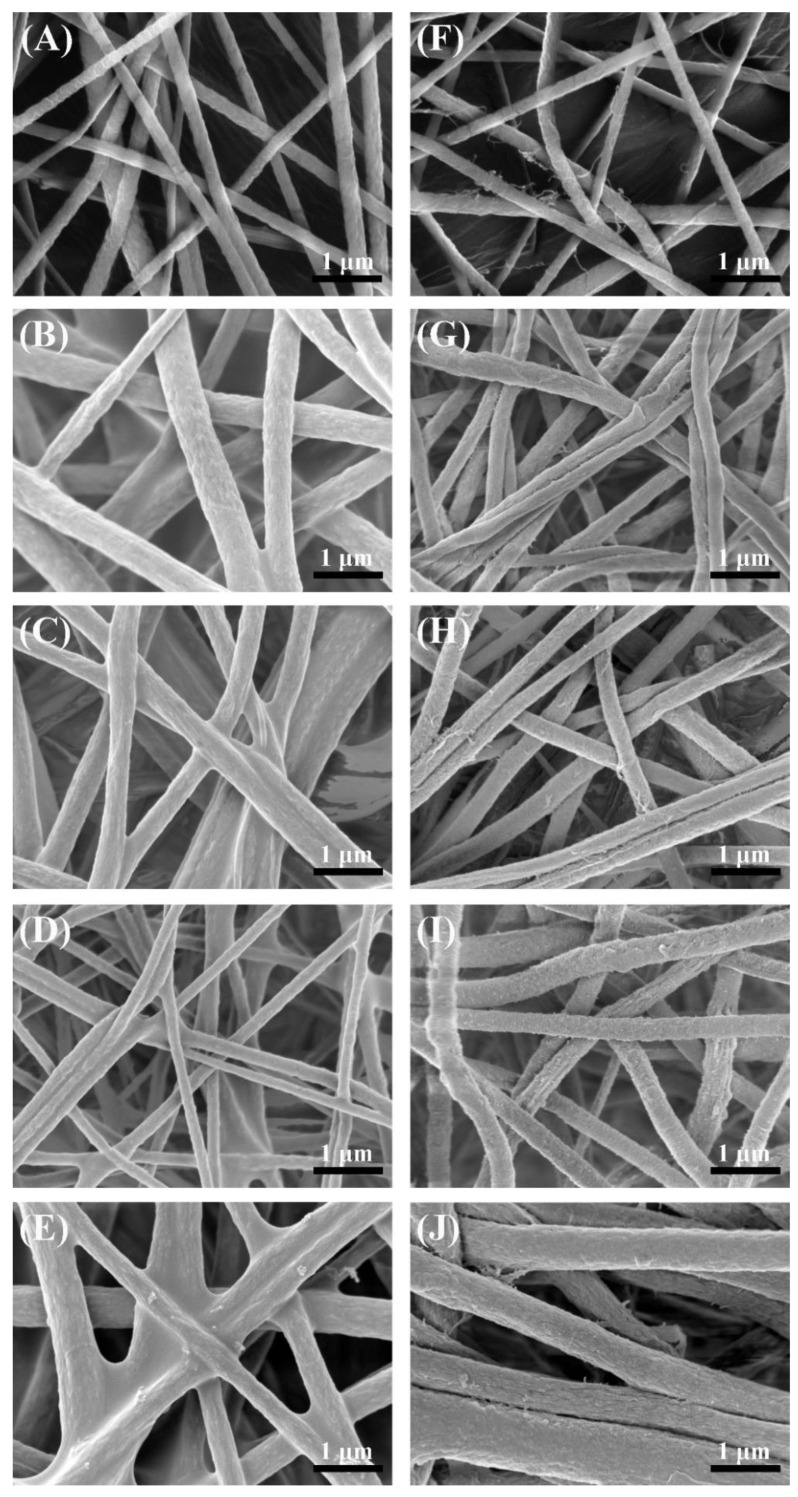
Morphology of the different membranes: (**A**) E-0; (**B**) E-1; (**C**) E-2; (**D**) E-3; (**E**) E-4; (**F**) ES-0; (**G**) ES-1; (**H**) ES-2; (**I**) ES-3; (**J**) ES-4.

**Figure 4 membranes-13-00225-f004:**
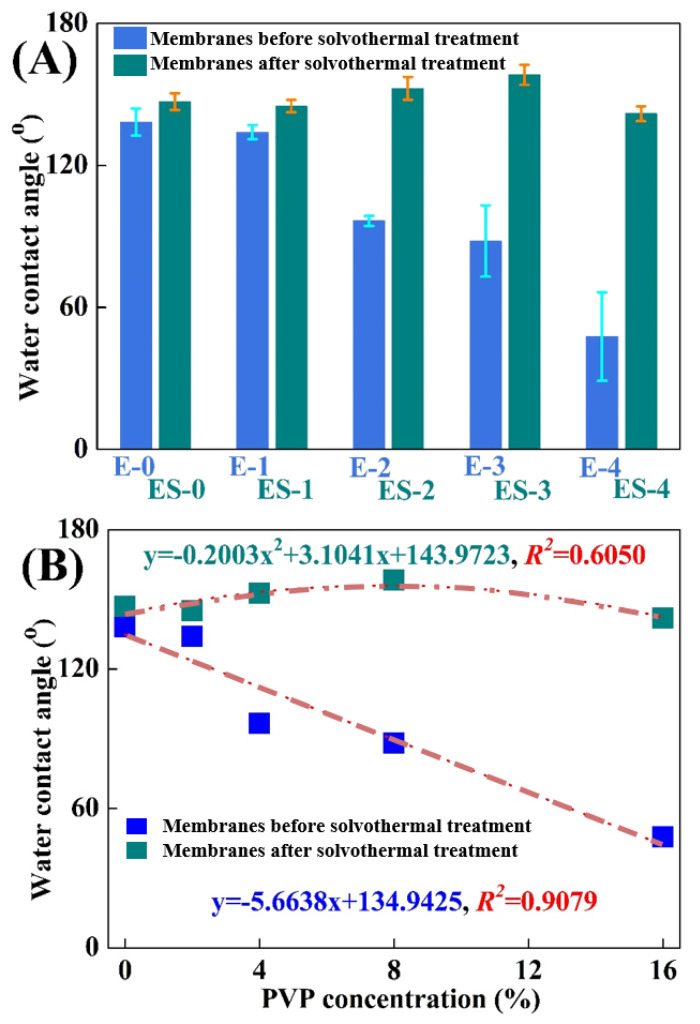
(**A**) Variations of the water contact angles on different membranes; (**B**) variations of the tendency of the water contact angles on different membranes.

**Figure 5 membranes-13-00225-f005:**
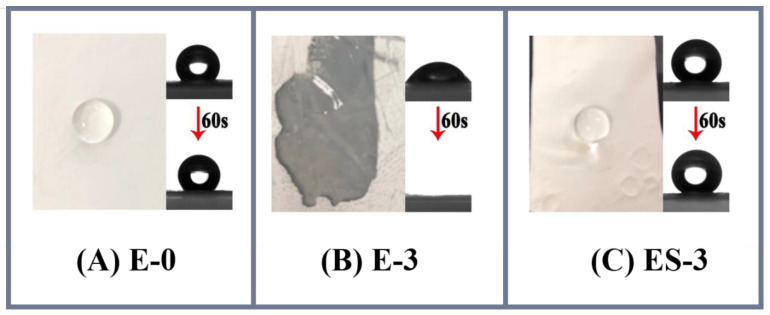
Membrane contact angle at the beginning and after 60 s: (**A**) E-0; (**B**) E-3; (**C**) ES-3.

**Figure 6 membranes-13-00225-f006:**
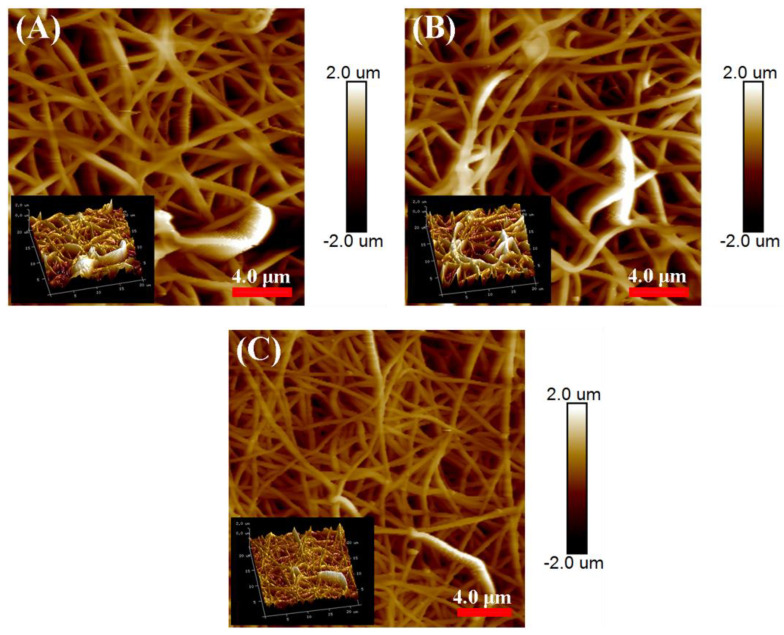
Roughness for the different membranes: (**A**) E-0; (**B**) E-3; (**C**) ES-3.

**Figure 7 membranes-13-00225-f007:**
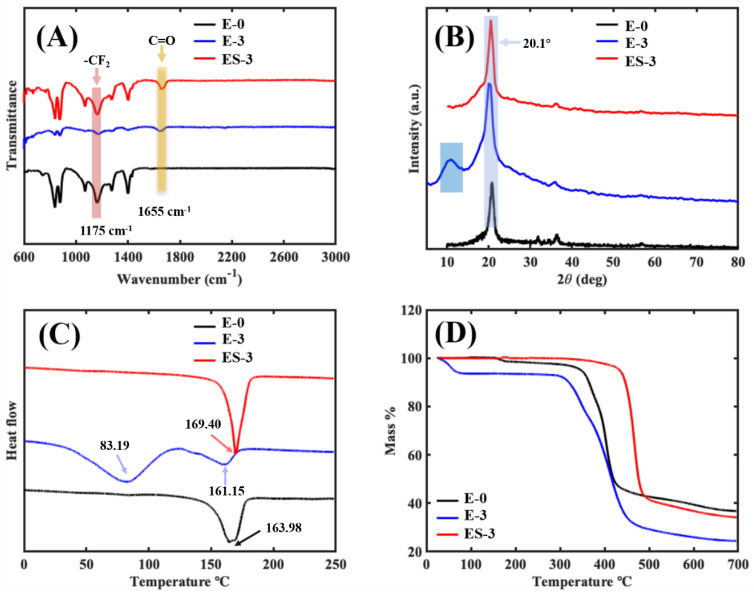
Characterization of the different membranes: (**A**) FTIR spectroscopy; (**B**) XRD spectrum; (**C**) DSC testing; (**D**) TGA analysis.

**Figure 8 membranes-13-00225-f008:**
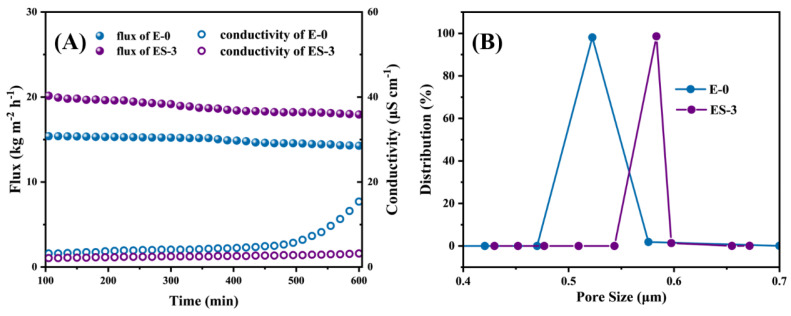
(**A**) MD performance of E-0 and ES-3 with NaCl solution without SDS; (**B**) pore size distribution of E-0 and ES-3.

**Figure 9 membranes-13-00225-f009:**
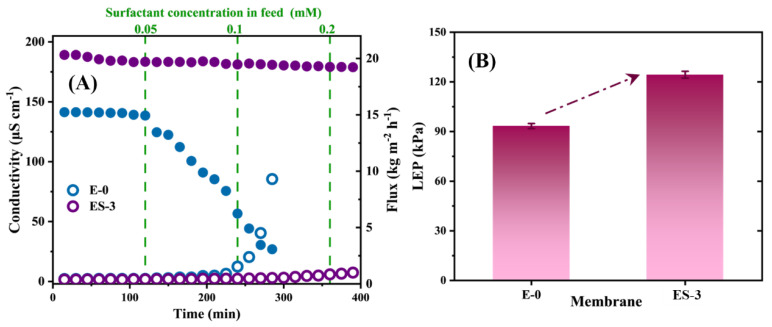
(**A**) MD performance of E-0 and ES-3 with SDS solution of different concentration; (**B**) LEP value of E-0 and ES-3.

**Table 1 membranes-13-00225-t001:** Components of the polymer solutions for the different membranes.

Samples	Polymer Solution			
	PVDF (%)	PVP (%)	DMAc (%)	Acetone (%)
E-0	8.0	0.0	73.6	18.4
E-1	8.0	2.0	72.0	18.0
E-2	8.0	4.0	70.4	17.6
E-3	8.0	8.0	67.2	16.8
E-4	8.0	16.0	60.8	15.2

**Table 2 membranes-13-00225-t002:** Properties of the different membranes.

Samples	*R_a_* (nm)	*R_q_* (nm)	Porosity (%)	Mean Pore Size (μm)
E-0	336	435	80.2	0.479
E-3	556	727	76.3	0.453
ES-3	598	764	83.5	0.561

**Table 3 membranes-13-00225-t003:** Comparison of the MD performances of the different membranes.

Membranes	Temperature Difference (°C)	Water Flux (LMH)	Salt Rejection (%)	SDS Concentration (mM)	Reference
PL-PVDF	50	7.95	99.9	-	[34]
F-TNF	60	12.2	>99.92	-	[35]
PVDF/PDMS	50	16.2	>99.98	0.3	[36]
Pf_w_-HNT	40	11.26	99.97	-	[37]
PET ENMs	40	19.13	99.9	0.2	[38]
ES-3	40	21.9	>99.98	0.2	This work

## Data Availability

The data presented in this study are available from the corresponding author upon reasonable request.

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
