# Peer review of "Anti-Wetting Performance of an Electrospun PVDF/PVP Membrane Modified by Solvothermal Treatment in Membrane Distillation"

_membranes, 2023, doi:10.3390/membranes13020225_

Round 1

Reviewer 1 Report

The publication shows sufficient new and interesting results, especially concerning the modification of Electrospun PVDF/PVP Membrane by Solvothermal Treatment in Membrane Distillation. The language need to be modified. However, there are some drawbacks of the paper due to the inclusion of "Chemical" part in it, which is not well described. Therefore, I suggest that the authors should have a major revision and collation of the article. For more details see the comments bellow.

Q1. In the abstract section the first two sentences need to be modified.

Q2. In Materials section, please provide the molecular weight and purity of all chemicals used in this work.

Q3. Is the viscosity of the Electrospinning solutions was measured? Please mesure it and supported the SEM (line 175 page 5)results with it.

Q4.  Page 3 line127, please add the concentration of HCL used in Solvothermal Treatment .

Q5. Fig 4 (A and B). Why authors don’t label the diagram ( pale blue and dark blue)?

Q6 . Fig. 6 . please calculate the prosity and pore size for E-0, E-3 and ES-3 membranes.

Q7. Fig. 7 (c ). Please calculate the crystalinity of different prepared membranes.

Q8. Q10. Make a comparison between your work and other previous works.

Q9. The conclusion part needs modification.

Reviewer 2 Report

This manuscript discusses the solvothermal treatment of PVDF/PVP membranes to improve the anti-wetting properties of membranes.  In this regard, I have some comments that should be addressed in order for the manuscript could be considered for publication.

1. Line 18: word virgin should be changed to native. Native seems to be more scientific.

2. Basic description of the solvothermal methods should be added to the introduction part of the manuscript.

3. Color of the bars/dots/lines should be changed. On the white and white print, the difference is not visible.

4. FTIR analysis: According to the Authors of this manuscript the molecular chain is flipped and changed. Therefore, a more detailed analysis of the FTIR spectra is needed. Did the intensity of bands of α and β increase or decrease?

5. Line 315: Did the first mass loss of E-3 may be related to the moisture?

6. There is no research/information related to the membrane's mechanical properties before and after the solvothermal treatment. This information should be added to the manuscript.

I am happy to recommend your manuscript for publication after minor changes.

Round 2

Reviewer 1 Report

The authors made all recommendations and the manuscript is suitable for publication in its present form in Membranes Journal.